# Quantifying T Cell Cross-Reactivity: Influenza and Coronaviruses

**DOI:** 10.3390/v13091786

**Published:** 2021-09-07

**Authors:** Jessica Ann Gaevert, Daniel Luque Duque, Grant Lythe, Carmen Molina-París, Paul Glyndwr Thomas

**Affiliations:** 1Department of Immunology, St. Jude Children’s Research Hospital, Memphis, TN 38105, USA; jessica.gaevert@stjude.org; 2St. Jude Graduate School of Biomedical Sciences, Memphis, TN 38105, USA; 3Department of Applied Mathematics, School of Mathematics, University of Leeds, Leeds LS2 9JT, UK; mmdfld@leeds.ac.uk (D.L.D.); grant@maths.leeds.ac.uk (G.L.); 4T-6, Theoretical Biology and Biophysics, Theoretical Division, Los Alamos National Laboratory, Los Alamos, NM 87545, USA

**Keywords:** cross-reactivity, pre-existing immunity, heterologous infection, mathematical modeling, competition process, bipartite network

## Abstract

If viral strains are sufficiently similar in their immunodominant epitopes, then populations of cross-reactive T cells may be boosted by exposure to one strain and provide protection against infection by another at a later date. This type of pre-existing immunity may be important in the adaptive immune response to influenza and to coronaviruses. Patterns of recognition of epitopes by T cell clonotypes (a set of cells sharing the same T cell receptor) are represented as edges on a bipartite network. We describe different methods of constructing bipartite networks that exhibit cross-reactivity, and the dynamics of the T cell repertoire in conditions of homeostasis, infection and re-infection. Cross-reactivity may arise simply by chance, or because immunodominant epitopes of different strains are structurally similar. We introduce a circular space of epitopes, so that T cell cross-reactivity is a quantitative measure of the overlap between clonotypes that recognize similar (that is, close in epitope space) epitopes.

## 1. Introduction

T cell receptors (TCRs) recognize peptides (or epitopes), typically 8–14 amino acids long, bound to major histocompatibility complex (MHC) class I and class II molecules on antigen-presenting cells. There cannot only exist a single TCR which recognizes a given peptide because the possible number of peptides is far greater than the number of T cells in a mouse (about $10^{8}$) or in one person (about $10^{12}$). For example, the set of 11-mers alone, assuming that three percent of them can be presented by MHC, amounts to $6\times 10^{12}$ different peptides [1]. Therefore, individual TCRs must recognize multiple peptides if a mammal’s T cell repertoire is to be capable of providing coverage against the majority of new pathogens a host might encounter in its lifetime. In other words, based on these arguments, made over twenty years ago by Don Mason, T cells must be cross-reactive [1]. More recently, it has become possible to examine the other side of cross-reactivity: in one mouse, how many different T cells, and how many different TCRs, recognize a given peptide? Peptide-MHC multimers can be used to count, in a mouse, the subset of T cells with specificity to a single peptide bound to MHC (pMHC) [2,3]. A few dozen to a few thousand cells have been found per epitope, and the epitope-specific subset is almost maximally diverse: most TCRs are found on only one cell in a mouse [4,5,6].

TCR cross-reactivity has become a key focus of many scientists in light of the SARS-CoV-2 pandemic and the potential for pandemic influenza [7,8,9,10,11,12]. Cross-reactive CD8${}^{+}$ T cells have the potential to be protective across heterologous infections [10], but little is known about the biology or dynamics of the cross-reactive response by CD8${}^{+}$ T cells. In this review, we aimed to survey both current biological and mathematical literature on CD8${}^{+}$ T cell cross-reactivity and pre-existing immunity, and to propose our own mathematical models of cross-reactivity in an immune system faced with heterologous viral infections. We focus on cross-reactive CD8${}^{+}$ T cells in the context of influenza and coronavirus infections. Both Andrew Sewell and Don Mason laid out the argument that T cells must be cross-reactive because the number of possible pMHCs that the immune system could encounter far exceeds the number of potential T cells in a given host [1,13]. In Mason’s paper their arguments are supported by the simple arithmetic that the weight of $10^{15}$ individual T cells (one per potential foreign peptide) would be greater than 500 kg [1]. Sewell argues that each T cell can recognize many different pMHCs [13]: "This compromise on specificity has profound implications because the chance of any natural peptide–MHC ligand being an optimal fit for its cognate TCR is small, as there will almost always be more-potent agonists" [13]. This view of TCR cross-reactivity provides both a plausible cause for autoimmune disease and a rationale for therapeutic interventions [13].

However, there is still little primary data aimed at understanding the molecular basis of cross-reactivity. Cross-reactivity could be an important instrument the immune system makes use of to combat rapidly evolving pathogens, such as influenza and coronaviruses [1,7,13]. Cross-reactivity of T cells is also particularly relevant in the context of pre-existing immunity and vaccination [10,14]. In fact, the immune system may exploit cross-reactive T cells to combat the diverse array of pathogens and individual encounters over a lifetime [1,13,15,16].

Mathematical modeling can be used as a quantitative tool to generate and test hypotheses about the mechanisms and rules of TCR and pMHC recognition. We propose, in what follows, a mathematical perspective to investigate cross-reactivity in T cells [17]. In the second part of our review, we make use of mathematical modeling to hypothesize about different mechanisms of CD8${}^{+}$ T cell cross-reactivity. For instance, we can think of cross-reactivity as being focused or unfocused; i.e., T cell clonotypes that are reactive to a given peptide are also cross-reactive with each other for other peptides, or they are cross-reactive with random clonotypes [1]. To consider CD8${}^{+}$ T cell cross-reactivity, we need to define the key terms of cross-reactivity, pre-existing immunity, heterologous infection and immunity. Cross-reactivity often has many uses and definitions within the literature, so for the purposes of this review, cross-reactive CD8${}^{+}$ T cells are those whose TCRs can recognize more than one epitope presented in the context of MHC (see Figure 1).

In this review, pre-existing immunity is defined as the immunity an individual has gained to a certain pathogen, and which has been previously built up by prior infections or vaccinations [10]. Heterologous immunity and heterologous infection also have many definitions within the literature [10,18]. We define heterologous immunity as immunity to multiple viruses or many different strains of a virus, such as influenza [19]. We are considering a heterologous infection with multiple viruses, either different strains of a single virus or different viruses altogether (see Figure 2). Pre-existing immunity generated from different viruses has become especially relevant in the current SARS-CoV-2 pandemic. Of particular relevance in this instance is the potential role played by pre-existing immunity in the form of cross-reactive T cells or B cells generated during previous common coronavirus infections [11,20,21]. Cross-reactive T cells can also arise from multiple influenza infections, and the primary infection can directly alter the immunodominance hierarchy observed in subsequent influenza infections [10].

In the context of one TCR fitting one pMHC, the number of possible TCRs that can be generated in the thymus far outnumbers the number of T cells a single immune system can sustain [1,22]. Antigen diversity is greater still than TCR diversity [1], and thus, in order for the immune system to be able to combat all pathogens it will see in a lifetime, cross-reactive T cells must exist and be functional [13]. More recently, cross-reactive T cells have been identified to play a role in the immune responses to several different viral infections, including coronaviruses (especially SARS-CoV-2), influenza viruses and HIV [10,11,20,23]. Previous viral exposure also plays a key part in cross-reactivity and the generation of immune memory. A key question to pose is, "What is the role of cross-reactivity in the infection history of an individual and how does this history, in turn, affect cross-reactivity?" While there is currently little data to answer this question, we can begin to test hypotheses while making use of mathematical modeling.

A quantitative approach will be particularly useful when elucidating details of cross-reactivity and its role in heterologous and pre-existing immunity [24,25,26]. From as far back as 1968, mathematical models have been used to describe the behavior of influenza viruses at a population level. In an early example, Baroyan, Rvachev et al. proposed a deterministic mathematical model to simulate influenza outbreaks across the 43 largest cities in the USSR [27,28]. In addition to reproducing observations and fitting models to real-life datasets, we can use mathematical models to propose and test new hypotheses, and then test these against new biological data (see Figure 3).

## 2. T Cell Cross-Reactivity in the Context of Influenza Viruses

One pathogen of great human relevance and importance to cross-reactivity is influenza [7,10,14,29]. Influenza viruses may elicit cross-reactive T cells because they exhibit antigenic shift and drift, and because individuals may suffer many influenza infections over a lifetime [7,30]. Antigenic drift takes place when the viral genome accumulates mutations introduced during error-prone replication over time [30] (see Figure 4). It is estimated that the error rate for the influenza RNA polymerase is between 10${}^{-4}$ and 10${}^{-6}$ substitutions per nucleotide per cell infection (approximately one error per replicated genome) and that nucleotide substitutions are much more common among RNA polymerase errors than deletions [31,32]. In contrast, antigenic shift occurs in the context of viral co-infection, due to viral segment reassortment, where entire sections of the segmented RNA genome are shifted in/out of the genome [30].

Several groups have identified potentially cross-reactive T cells elicited in response to different influenza viruses and have directly applied this data to immunodominance hierarchies in influenza [10,14,29,33,34]. Immunodominance is defined as the phenomenon in which immune responses are only mounted against a few of the antigenic peptides out of the many potentially generated by the pathogen [10,14,29,34,35,36]. Duan et al., in 2015, were able to show that cross-reactive CD8${}^{+}$ T cells elicited during a primary infection can be recalled during a secondary heterologous influenza infection and effectively alter the immunodominance hierarchy [10]. The immunity generated by a previous influenza infection in this case can be considered “pre-existing immunity” when the mice, later challenged with the more severe H7N9 strain, were protected [10]. In addition, Duan et al. predicted that many T cell epitopes from pandemic influenza in 1968 would be cross-reactive with the pandemic H1N1 strain of 2009 and that this cross-reactivity would be between immunodominant epitopes [10].

In another report, the authors presented immunological and structural analyses of cross-reactive CD8${}^{+}$ T cell-mediated immunity directed at the variable, but highly cross-reactive, immunodominant NP_418_ peptide that binds to the B7 family (HLA-B*3501/03/0702), widely expressed in human populations [14]. They found that specific amino acid residues within the NP_418_ peptide play crucial roles in cross-reactivity and T cell responses induced by this epitope [14]. This is based on the fact that many of the immunogenic peptides derived from the H1N1 2009 influenza virus are representative of the 1918 influenza pandemic strain, rather than of more recent seasonal strains [14]. Kedzierska et al. also investigated potential CD8${}^{+}$ T cell cross-reactivity among influenza epitopes in 2009 and found that some epitopes could elicit a cross-reactive response (detected via cytokine production in an ELISpot assay) in cell cultures [29]. From these studies one can infer that cross-reactivity and immunodominance are explicitly linked. However, the precise impact of cross-reactivity on immunodominance and memory T cell responses seems to be priming strain specific [10]. This adds another association between cross-reactivity and reassortment, since reassorted genomes can both help the immune response or allow for viral escape.

Modeling host–pathogen dynamics, including viral replication and CD8${}^{+}$ T cell activation, expansion and contraction, with ordinary differential equations (ODEs), such as the ones used by Baccam et al., implicitly assumes that the time a cell spends in its infectious stage (infected with no virion shedding and infected with virion shedding) is exponentially distributed [37]. Holder and Beauchemin showed that in the case of influenza A infections, the use of exponential or fixed time delays results in virus concentrations that very poorly fit experimental data [38]. Fixed time delays provided a better fit only when the growth times were linear, but not during the early stages of infection [38]. They found that normal or lognormal distributed waiting times led to a better fit to the data over the entire range of experimental timescales [38]. The model proposed by Baccam et al. also considered immune responses to infection in the form of type I interferons (IFNs) secreted as part of the innate immune response [37,39,40,41]. This allowed the model to replicate a bimodal virus titre peak present in the experimental data [42,43,44]. This model could not be reproduced by only considering a limit on the number of target cells available [42,43,44]. However, the second peak of the viral titre obtained did not occur at the same timescale observed in the experimental data, suggesting that a more robust model that incorporates a wider immune response (e.g., T cells, antibodies, NK cells) could better explain the data [45].

Further work on this type of model is still required to better understand the dynamics of immune responses to influenza A viruses [45]. A mathematical model considering the innate immune response to influenza viruses was proposed by Cao, et al., to explain viral hierarchies in the context of multi-strain infections [46]. However, while models where the adaptive immune response is included have been developed for other viruses, such as the West Nile virus, this work has not yet been extended to influenza viruses [47]. Influenza serves as our first example of a human viral infection which can lead to pre-existing cross-reactive immunity. Since T cell cross-reactivity is not completely understood, we put forward our hypotheses of how cross-reactive immunity might work in Section 4, Section 5 and Section 6 of this review [1,13,19,48].

## 3. T Cell Cross-Reactivity in the Context of Coronaviruses Infection

Cross-reactivity plays an important role in immunity to SARS-CoV-2, the coronavirus responsible for the COVID-19 pandemic. Similar to influenza viruses, coronaviruses infect many different animals and humans, and can cause mild to severe respiratory infections [9,49]. However, coronaviruses differ significantly in their replication and genome composition. All coronaviruses are enveloped, positive, single stranded RNA viruses [49]. SARS-CoV-2 is a beta-coronavirus, as are severe acute respiratory syndrome (SARS-CoV-1) and Middle Eastern respiratory syndrome (MERS) [49] coronaviruses. Coronaviruses do not undergo genetic shift because they cannot reassort, but their genomes are subject to both homologous and non-homologous recombination [49,50] due to the unique aspects of their RNA-dependent RNA polymerase (RDRP), which likely plays a role in emergence of new coronaviruses and viral evolution [50]. Frame shift mutations are also common occurrences during replication of coronaviruses [50]. Previous exploration of cross-reactivity in coronaviruses indicates that the similarities between the Mn2 peptide in SARS-CoV-1 and its parallel peptide in MERS may elicit cross-reactive T cells [9]. However, this observation has not been experimentally validated [9]. Long term T cell immunity to SARS-CoV-1 and MERS have also been found but not experimentally explored further [51]. For SARS-CoV-1 there was a study looking for cross-reactivity with common cold coronaviruses (CCCs) [52]; yet, this group did not find any cross-reactivity between CCCs and SARs-CoV-1 [52].

However, since the onset of the current pandemic, pre-existing immunity potentially generated by these viruses and the existence of cross-reactive T cells have become the key interests of several research groups [11,20,21]. In particular, these groups are now exploring the possibility of cross-reactivity between CCCs and SARS-CoV-2—specifically, the possibility of pre-existing immunity to SARS-CoV-2 as a result of seasonal exposure to CCCs [11,20,21]. In these papers by Grifoni et al., CD4${}^{+}$ T cells cross-reactive between CCCs and SARS-CoV-2 were identified using peptide pooling ELISpot assays to look for functional responses by the T cells [11,20]. Mateus et al. also asserted that pre-existing immunity to SARS-CoV-2 is due to the presence of cross-reactive T cells elicited from prior infections with CCCs [21]. They found CD4${}^{+}$ T cells which recognize epitopes derived from the spike proteins of SARS-CoV-2 and CCCs OC43, NL63, HKU-1, and 229E [21]. However, this paper focuses exclusively on the cross-reactive potential and repertoire of CD4${}^{+}$ T cells, but not on the existence or role of cross-reactive CD8^+^ T cells [21].

Using pools of peptides from CCCs and SARS-CoV-2 in conjunction with ELISpot and activation induced marker (AIM) assays, the authors were able to identify cross-reactive CD4^+^ T cells to homologous epitopes in the receptor binding domain (RBD) and other spike regions of these viruses [21]. In addition, Richards et al. also found that circulating CD4^+^ T cells originally elicited in response to CCCs can functionally respond to SARS-CoV-2 via ELISpots for IL-2, granzyme-B and IFN-γ [53]. However these responses were extremely variable and decreased with age [53]. If there was a robust cross-reactive response, they would have expected to not see a decrease in circulating memory CD4${}^{+}$ T cells over time [53]. Specifically, the largest cross-reactive effect they observed was in response to peptide stimulation with the spike RBD of CCCs [21]. This is not to say that cross-reactive CD8${}^{+}$ T cells across SARS-CoV-2 and CCCs do not exist, just that those studies may be ongoing and yet to be published. Connecting the findings of cross-reactive CD4${}^{+}$ T cells and pre-existing immunity generated from previous CCCs, cross-reactive CD8${}^{+}$ T cells very likely play a role in the adaptive immune response to SARS-CoV-2 [20,21,54] (see Figure 2 and Figure 5). Additionally, Lee et al. were able to use in silico methods to profile public and private epitopes of SARS-CoV-2 infections [55]. They concluded that there is potential for cross-reactive CD8${}^{+}$ T cells elicited from CCCs that result in pre-existing immunity to SARS-CoV-2. However, from their results, they predicted that the number of cross-reactive epitopes is relatively small [55]. Recently, Saini et al. published their results showing increased CD8${}^{+}$ T cell activation in SARS-CoV-2 patients [54]. Using pMHC DNA barcoded multimers in combination with a T cell phenotyping panel, they were able to identify a set of 122 immunogenic epitopes and a subset of immunodominant SARS-CoV-2 T cell epitopes [54]. They found that up to 27% of the CD8${}^{+}$ T cells in an individual could react to SARS-CoV-2 epitopes [54]. In thinking about these datasets, given the large magnitude of the T cell response, a role for cross-reactivity in the adaptive immune response to SARS-CoV-2 seems likely (see Figure 5).

While it is yet to be definitively shown how CD8${}^{+}$ T cells play a role in cross-reactivity and pre-existing immunity to SARS-CoV-2, other coronavirus and influenza infections, we were able to propose and mathematically model different hypotheses of CD8${}^{+}$ T cell cross-reactivity and evaluate their relevance to the dynamics of immune responses to SARS-CoV-2 (or other pathogens), making use of quantitative methods.

Circling back to the ideas of pre-existing immunity and heterologous infections, the immunity to SARS-CoV-2 generated from previous CCC infections exemplifies the potential for cross-reactivity in the current pandemic [11,20,53,54]. The immunity generated by these CCCs prior to SARS-CoV-2 infection is in this case pre-existing immunity [11,20,53,54] (see Figure 2 and Figure 5). This pre-existing immunity is then re-activated upon exposure to the heterologous SARS-CoV-2 infection [11,20,53,54]. In addition, this was also observed by Duan et al., where priming with various heterologous influenza viruses was protective upon secondary challenge with the more severe H7N9 influenza strain in mice [10]. However, in this particular set of experiments it was noticed that the protective effect from T cells and the changes in the immunodominance hierarchy of epitopes were directly related to the priming infection [10]. In both of these contexts, we see how important cross-reactive T cells can potentially be in situations where the immune system relies on pre-existing immunity to combat a current heterologous infection.

## 4. Modeling T Cell Cross-Reactivity with a Bipartite Recognition Network

We were interested in developing mathematical models to describe the dynamics of T cell responses to different pathogens, and T cell cross-reactivity and its roles in infection, T cell homeostasis and the establishment of T cell memory. We followed the mathematical methods and approaches developed by Stirk et al. [56,57,58,59,60] and introduced a bipartite network which encodes the pMHC (or antigen) recognition pattern for each T cell clonotype, characterized by its specific TCR (see Figure 6). We note that the bipartite network introduced by Stirk et al. considers only self-pMHC complexes. In this review, we generalize the bipartite recognition network to allow for the consideration of foreign antigens being presented. In a bipartite (recognition) network there exist two types of nodes. Network edges only connect nodes of a different kind. For our purposes the two types of nodes are T cell clonotypes and peptides in the context of MHC presentation, respectively. An edge between a T cell clonotype and a peptide-MHC is drawn (or exists) if the given TCR clonotype recognizes the chosen peptide-MHC. We classified the set of peptide-MHCs according to whether they are derived from house-keeping proteins (or self), and we call them self-peptide-MHCs (self-pMHCs) [61,62], or whether they are virus-derived peptide-MHCs (VDPs) [63,64]. Figure 6 shows an example of a bipartite recognition network between T cell clonotypes and VDPs from two viruses, which will be later used in Section 5.1 to describe the dynamics of a T cell immune response to two heterologous viral infections.

In what follows we want to explore ways to make use of bipartite recognition networks between T cells and peptide-MHCs to model the dynamics of T cell clonotypes during a viral infection. To this end we assume that if a T cell clonotype and a VDP share an edge, then during an infection with that virus (from which the chosen VDP is derived) the clonotype will proliferate and expand. It thus follows that the bipartite recognition network (for self-pMHCs and VDPs) encodes the dynamics of T cell clonotypes and their population of cells in homeostasis and during infection. Before we do so, it is important to discuss potential ways to define a bipartite recognition network—that is, how to assign edges between T cell nodes and pMHCs, and how the network might encode our knowledge of existing cross-reactivity [1,13].

### 4.1. Constructing the TCR-pMHC Recognition Network

When constructing a TCR-pMHC recognition network, we assign to each clonotype which pMHCs (self or VDPs) it will be able to recognize. In what follows and since our primary focus is infection, in this review we restrict ourselves to VDPs, with the understanding that the same principles can be easily generalized to consider self-pMHCs [56,65,66]. We can visualize this construction as follows: imagine a bag of T cell clonotypes (the complete TCR repertoire of an individual) and a bag of VDPs (belonging to the first and second infection), as shown in Figure 7. For each clonotype we draw a sample of VDPs that it will be able to recognize. Thus, the rules of how we draw such sample determine and define the bipartite recognition network for a given individual. By using different sampling strategies we can construct networks that represent focused or unfocused cross-reactivity [1,13]. In short, we aim in this section to formalize in a mathematical manner the cross-reactivity recognition ideas first introduced by Mason [1]. We limit our consideration to three different sampling strategies, and discuss how they relate to cross-reactivity.

#### 4.1.1. VDP Sampling for Unfocused Cross-Reactivity

The first case we want to consider is that of a bipartite recognition network with unfocused cross-reactivity [1]. Unfocused cross-reactivity, as defined in Ref. [1], means that any cross-reactivity present in the bipartite recognition network is due to chance; that is, it is random. Our proposed sampling strategy to generate such behavior is as follows: each VDP is equally likely to be extracted from the bag, and after a VDP sample is taken (for a chosen T cell clonotype), all VDPs are replaced back in the bag. In order to formalize the sampling strategy used to construct the VDP recognition network, we will make use of Bernoulli experiments.

Let us define *p* to be the probability that a VDP will be drawn from the bag, and therefore, 1−p is the probability that it will not be extracted. This probability is the same for all VDPs in the bag. If our bag contains a single VDP, the previous experiment of sampling is known as a Bernoulli experiment, with probability of success *p* (and probability of failure 1−p). Now, since we are considering a bag with not just one but many VDPs in it, we have a system of *M* independent Bernoulli experiments, with *M* the number of VDPs in the bag. A sum of Bernoulli experiments is modeled by a binomial distribution. Thus, if we consider X to be the random variable describing the number of VDPs that a clonotype can recognize, given that our bag has *M* different VDPs, then this random variable will follow a binomial distribution. We write X∼Binomial(M,p) and we call this network the unfocused cross-reactivity network.

#### 4.1.2. VDP Sampling for Focused Cross-Reactivity

The second case we want to consider is that of a bipartite recognition network with focused cross-reactivity [1]. We would like to explore, for the purposes of illustration, two different kinds of bipartite recognition network with focused cross-reactivity. We note that this exploration of focused cross-reactivity is not exhaustive. In fact, there exist many other different ways to generate bipartite networks with focused cross-reactivity. Given the limited space in this review, we restrict ourselves to only two such examples. A simple way to generate a focused cross-reactivity network is as follows: define a priori the number of VDPs each clonotype will recognize; that is, the number of edges which will be assigned to each clonotype is set to be a constant. The number of edges of a node in a network is called the degree of that node. We can think of this as choosing an integer number, *k*, of VDPs that we want every clonotype to recognize. For each clonotype we sample VDPs from the bag (with a Bernoulli experiment and probability *p* of success) until we reach a total of *k* different successes. The set of *k* successes is a set of *k* different VDPs, which should be assigned edges to the chosen clonotype in the bipartite network. In this case, every individual VDP is again a Bernoulli experiment. However, since we are interested in having exactly *k* successes for each choice of T cell clonotype, this sampling strategy is not a binomial distribution, but a negative binomial distribution. Thus, X, the random variable describing the number of VDPs that a clonotype can recognize, is distributed according to a negative binomial distribution, and we write X∼NB(k,p). This distribution models the number of trials before *k* successes occur, and each success has a probability *p*. Since this bipartite network is characterized by the degree, *k*, of its TCR nodes, we call it the fixed degree focused cross-reactivity network.

For the second type of focused cross-reactivity network, we hypothesized, following the argument presented by Mason [1], that when two clonotypes can recognize the same VDP, there is a greater probability that they will share other VDPs as well. For example, in Figure 8 shared recognition of $v_{2}$ between clonotypes *i* and *j* increases the probability of an edge between *j* and $v_{3}$ being added, since *i* recognizes $v_{3}$. (We note that in this case clonotype *i* was added to the network prior to clonotype *j*). In this way, this sampling strategy creates clusters of clonotypes that have similar recognition patterns. Following the Barabási-Albert model [67], we construct a second type of focused cross-reactivity network as follows. We chose a clonotype at random from the clonotype bag. Given a choice of clonotype, we then sampled the bag of VDPs, making use of the binomial distribution, as described in Section 4.1.1. Once we did this, we chose other clonotypes, randomly one at a time and without replacement, and sample from the bag of VDPs (for a given choice of T cell clonotype) in two steps: (i) we made use of the binomial distribution to sample the initial VDPs the clonotype would recognize. We then checked this sample to identify the VDPs that were being shared with clonotypes already included in the bipartite network; (ii) if there existed any such shared VDPs, we sampled from the set of VDPs that said clonotype(s) could recognize with the binomial distribution, and added them to the set of VDPs the new clonotype can recognize. We repeated this step for any shared VDPs. The process of adding edges based on already existing shared nodes is called preferential attachment, and was introduced in the Barabási-Albert models [67,68,69] for other purposes not related to immunology. We call this network the preferential attachment focused cross-reactivity network.

## 5. Dynamics of T Cells during a Viral Infection

A fundamental part of modeling the dynamics of a system is deciding which parameters we should use to define and characterize its dynamics. When deciding these parameters we aimed to choose only those that would allow us to accurately model the dynamics, while also answering the questions we ask. For this review, since we were interested in the population dynamics of the T cell repertoire, we focused on parameters that describe behaviors that directly affect the populations of cells. For example, how often do naive cells divide in homeostasis? And how quickly do effector cells die in the absence of stimulus? In Table 1 we have listed the most important parameters we identified, and the symbols we use to represent them.

### 5.1. Modeling with an Example: Two Heterologous Viral infections

Thus far, we have only considered and discussed the construction of the recognition network between T cell clonotypes and pMHCs, but have not addressed how a network can be used to mathematically model the population dynamics of T cell clonotypes during a viral infection. For the purposes of this review, and without lack of generality, let us consider the case of two heterologous viral infections. We assume the bipartite recognition network corresponding to this toy model is given by Figure 6. In our toy model of infection and T cell immune response, our scenario is an individual infected by two heterologous viruses (which do not share any VDPs), with a recovery period between them; that is, the individual is first infected with a virus characterized by VDPs from the set $V=\{v_{1},v_{2},v_{3}\}$, and later infected with a virus characterized by VDPs from the set $W=\{w_{1},w_{2},w_{3}\}$. We assume the individual is naive to each of these two viral infections and that their naive T cell repertoire consists of three different clones (see Figure 6). During each infection, naive T cells which recognize VDPs will differentiate to become effector cells, which once infection is cleared will contract and generate specific memory cells.

In Figure 9 we describe the differentiation pathway for naive, effector and memory pools, which will be used to define the mathematical model describing the dynamics of the T cell immune response [70,71]. *N*, *E* and *M* represent the naive, effector and memory pools, respectively. Black arrows represent homeostatic proliferation rates for naive and memory cells. Red arrows represent the death rates for all three populations. Blue arrows represent infection (or VDP)-mediated differentiation and division rates. The purple arrow represents the differentiation rate from effector cells into the memory pool. We assume memory to be formed mostly once the viral infection has been cleared, as only a negligible number of memory cells are generated during an infection from effector cells [63]. The rate of differentiation from effector to memory, $\psi_{E}$, was chosen so that $\frac{\psi_{E}}{\psi_{E}+\mu_{E}}=\beta$, with β a fraction in the range of 5−10% [63]. In this way, we set $\psi_{E}=\frac{\beta }{1-\beta }\mu_{E}$.

We now introduce the vector, n, which describes the T cell population of each phenotype (naive, effector, memory) for all three clonotypes. For our toy model, and as shown in Figure 6, we have
(1)$$n=(n_{1},n_{2},n_{3},e_{1},e_{2},e_{3},m_{1},m_{2},m_{3}),$$
where $n_{i}$ is the number (non-negative integer) of naive T cells of clonotype *i*, $e_{i}$ is the number of effector cells and $m_{i}$ is the number of memory cells, with i=1,2,3.

#### 5.1.1. Death Events

We assume death rates to be linear; that is, if we consider the state of our population at time *t* to be n as defined above, then the death rates for cells of clonotype *i* are given by
(2)$$\mu_{N}^{\left(i\right)}\left(n\right)=\mu_{N}\;n_{i},$$
(3)$$\mu_{E}^{\left(i\right)}\left(n\right)=\mu_{E}\;e_{i},$$
(4)$$\mu_{M}^{\left(i\right)}\left(n\right)=\mu_{M}\;m_{i},$$
where $\mu_{N}$, $\mu_{E}$ and $\mu_{M}$ are the per cell death rates for naive, effector and memory cells, respectively. We will assume
(5)$$\mu_{M}<\mu_{N}<\mu_{E}.$$

#### 5.1.2. Homeostatic Division-Events

For cells in the naive pool we assumed that homeostatic maintenance is ensured by self-pMHC presentation, which provides naive T cells with survival stimuli to divide [62,72,73]. If a naive cell receives a TCR-mediated signal from a VDP, in the context of a viral infection, this leads to naive T cell activation and differentiation into the effector T cell pool [63,74,75]. We assumed effector cells are not homeostatically maintained once the infection is cleared. Most effector cells will die, and a small fraction differentiates into memory cells. Thus, we expected the effector population to decline in the absence of pathogen. The memory population is homeostatically maintained separately from the naive or effector pools in a cytokine-mediated process [62,63,76,77].

The homeostatic division rate of naive cells was calculated, as initially proposed by Stirk et al. [56,57,58], by dividing the total amount of stimulus available to them per self-pMHC, which we assumed to be a constant, by the total number of naive cells that can recognize that self-pMHC complex [78,79,80,81]. As an illustrative example, consider the clonotypes in Figure 6, and denote by $\varphi_{i}$ the total homeostatic division stimulus from self-pMHCs available to clonotype *i*, with i=1,2,3. If at time *t* the population is in state n, defined in Equation (Equation 1), the homeostatic division rates are given by
(6)$$\sigma_{N}^{\left(1\right)}\left(n\right)=\varphi_{1}\;n_{1}\left(\frac{p_{1}}{n_{1}}+\frac{p_{12}}{n_{1}+n_{2}}+\frac{p_{13}}{n_{1}+n_{3}}+\frac{p_{123}}{n_{1}+n_{2}+n_{3}}\right),$$
(7)$$\sigma_{N}^{\left(2\right)}\left(n\right)=\varphi_{2}\;n_{2}\left(\frac{p_{2}}{n_{2}}+\frac{p_{21}}{n_{2}+n_{1}}+\frac{p_{23}}{n_{2}+n_{3}}+\frac{p_{231}}{n_{2}+n_{1}+n_{3}}\right),$$
(8)$$\sigma_{N}^{\left(3\right)}\left(n\right)=\varphi_{3}\;n_{3}\left(\frac{p_{3}}{n_{3}}+\frac{p_{31}}{n_{3}+n_{1}}+\frac{p_{32}}{n_{3}+n_{2}}+\frac{p_{312}}{n_{3}+n_{1}+n_{2}}\right),$$
where $p_{i}$ (i=1,2,3) is the the probability that clonotype *i* does not share self-pMHCs with clonotypes *j* and *k* (with j≠i≠k); $p_{ij}$ (i,j=1,2,3 with i≠j) is the probability that self-pMHCs recognized by clonotype *i* are also recognized by clonotype *j* with i≠j; and $p_{ijk}$ (i,j,k=1,2,3 with i≠j, i≠k and j≠k) is the probability that self-pMHCs recognized by clonotype *i* are also recognized by clonotypes *j* and *k*. It can be shown that these probabilities obey the following relationships for any clonotypes *i*, *j*, and *k* with i≠j, i≠k and j≠k [57]:$$\begin{array}{c} \varphi_{i}p_{ijk}=\varphi_{j}p_{jki}=\varphi_{k}p_{kij},\\ \varphi_{i}p_{ij}=\varphi_{j}p_{ji},\\ p_{i}+p_{ij}+p_{ik}+p_{ijk}=1. \end{array}$$

Memory cell homeostasis is regulated by cytokine signals, and thus, we assume for i=1,2,3.
(9)$$\sigma_{M}^{\left(i\right)}\left(n\right)=\sigma_{M}m_{i}\;.$$

The previous equation implies that the rate of homeostatic division per memory cell, $\sigma_{M}$, does not depend on its TCR specificity. This is in line with our assumption that memory T cells are homeostatically regulated by cytokine signals, such as IL-7 or IL-15 [62,63,76,77].

#### 5.1.3. Infection-Induced Differentiation and Division Events

We now consider the events (differentiation and division) that are induced by the viral infection. To do so we must describe the T cell dynamics associated with the bipartite network shown in Figure 6. The network indicates that different clonotypes compete for VDPs, and thus, we need to use the sets $V_{i}$, $W_{i}$, $C^{v}$ and $C^{w}$ to determine exactly which VDPs are recognized by a given clonotype, and which other clonotypes can also recognize those VDPs. Let us again consider the clonotypes presented in Figure 6. We assume the population of T cells at time *t* is in state n. We propose to model the per cell stimulus provided by VDPs as follows. For clonotype 1, the per cell stimuli from VDPs during the first and second infection are, respectively, given by
(10)$$\delta_{V}^{\left(1\right)}\left(n\right)=\frac{\gamma (v_{2})}{\left(n_{1}+e_{1}+m_{1}\right)+\left(n_{2}+e_{2}+m_{2}\right)+\left(n_{3}+e_{3}+m_{3}\right)}+\frac{\gamma (v_{3})}{n_{1}+e_{1}+m_{1}},$$
(11)$$\delta_{W}^{\left(1\right)}\left(n\right)=\frac{\gamma (w_{1})}{\left(n_{1}+e_{1}+m_{1}\right)+\left(n_{2}+e_{2}+m_{2}\right)}+\frac{\gamma (w_{2})}{n_{1}+e_{1}+m_{1}},$$
where $\gamma (v_{k})$ is the total stimulus from VDP $v_{k}$ and $\gamma (w_{k})$ is the total stimulus from VDP $w_{k}$, for k=1,2,3. For clonotype 2 the per cell stimuli provided by VDPs during the first and second infections, respectively, are given by
(12)$$\delta_{V}^{\left(2\right)}\left(n\right)=\frac{\gamma (v_{2})}{\left(n_{2}+e_{2}+m_{2}\right)+\left(n_{1}+e_{1}+m_{1}\right)+\left(n_{3}+e_{3}+m_{3}\right)},$$
(13)$$\delta_{W}^{\left(2\right)}\left(n\right)=\frac{\gamma (w_{1})}{\left(n_{2}+e_{2}+m_{2}\right)+\left(n_{1}+e_{1}+m_{1}\right)}+\frac{\gamma (w_{3})}{n_{2}+e_{2}+m_{2}}.$$

Finally, clonotype 3 only recognizes VDPs of the first infection, and its per cell stimulus can be written as
(14)$$\delta_{V}^{\left(3\right)}\left(n\right)=\frac{\gamma (v_{2})}{\left(n_{3}+e_{3}+m_{3}\right)+\left(n_{1}+e_{1}+m_{1}\right)+\left(n_{2}+e_{2}+m_{2}\right)}+\frac{\gamma (v_{1})}{n_{3}+e_{3}+m_{3}}.$$

We note that $\delta_{W}^{\left(3\right)}\left(n\right)=0$. We have just defined the per cell VDP stimulus rate for a given clonotype and a given viral infection. We have also assumed that at time *t* the T cell population is in state n, as defined in Equation (Equation 1); that is, $n=(n_{1},n_{2},n_{3},e_{1},e_{2},e_{3},m_{1},m_{2}$ and $m_{3}).$ This means that at time *t*, for instance, there are $n_{1}$ naive cells of clonotype 1, $e_{2}$ effector cells of clonotype 2 and $m_{3}$ memory cells of clonotype 3. We are now ready to introduce the rates that characterize our new stochastic model for the differentiation and division events induced by the viral infection.

Differentiation of naive to effector cells during infection (V or W): for a given clonotype *i* we multiply the per cell stimulus defined above, $\delta_{V}^{\left(i\right)}\left(n\right)$ or $\delta_{W}^{\left(i\right)}\left(n\right)$, by the number of naive cells of the clonotype, $n_{i}$, and by the per cell differentiation rate of naive to effector cells, $\alpha_{N}$.Differentiation of memory to effector cells during infection (V or W): for a given clonotype *i* we multiply the per cell stimulus defined above, $\delta_{V}^{\left(i\right)}\left(n\right)$ or $\delta_{W}^{\left(i\right)}\left(n\right)$, by the number of memory cells of the clonotype, $m_{i}$, and by the per cell differentiation rate of memory to effector cells, $\alpha_{M}$.Proliferation of effector cells during infection (V or W): for a given clonotype *i* we multiply the per cell stimulus defined above, $\delta_{V}^{\left(i\right)}\left(n\right)$ or $\delta_{W}^{\left(i\right)}\left(n\right)$, by the number of effector cells of the clonotype, $e_{i}$, and by the per (effector) cell proliferation rate, $\lambda_{E}$.Differentiation of effector to memory cells once the infection has been cleared: for a given clonotype, we assume that once the infection has been cleared, effector T cells become memory with a per cell rate $\psi_{E}$, which does not depend on the T cell clonotype (or TCR).

We can then write the differentiation rates of naive and memory cells of clonotype *i* as
(15)$$\alpha_{N}^{\left(i\right)}\left(n\right)=\left\{\begin{array}{cc} \alpha_{N}\;n_{i}\;\delta_{V}^{\left(i\right)}\left(n\right) & \mathrm{for}\;\mathrm{the}\;\mathrm{first}\;\mathrm{infection},\\ \alpha_{N}\;n_{i}\;\delta_{W}^{\left(i\right)}\left(n\right) & \mathrm{for}\;\mathrm{the}\;\mathrm{second}\;\mathrm{infection}, \end{array}\right.$$
(16)$$\alpha_{M}^{\left(i\right)}\left(n\right)=\left\{\begin{array}{cc} \alpha_{M}\;m_{i}\delta_{V}^{\left(i\right)}\left(n\right) & \mathrm{for}\;\mathrm{the}\;\mathrm{first}\;\mathrm{infection},\\ \alpha_{M}\;m_{i}\delta_{W}^{\left(i\right)}\left(n\right) & \mathrm{for}\;\mathrm{the}\;\mathrm{second}\;\mathrm{infection}, \end{array}\right.$$
respectively. The division rate of effector cells of clonotype *i* is
(17)$$\lambda_{E}^{\left(i\right)}\left(n\right)=\left\{\begin{array}{cc} \lambda_{E}\;e_{i}\;\delta_{V}^{\left(i\right)}\left(n\right) & \mathrm{for}\;\mathrm{the}\;\mathrm{first}\;\mathrm{infection},\\ \lambda_{E}\;e_{i}\;\delta_{W}^{\left(i\right)}\left(n\right) & \mathrm{for}\;\mathrm{the}\;\mathrm{second}\;\mathrm{infection}, \end{array}\right.$$
and the differentiation rate of effector cells to memory cells is
(18)$$\psi_{E}^{\left(i\right)}\left(n\right)=\psi_{E}\;e_{i}\;,$$
with $\psi_{e}=\frac{\beta }{1-\beta }\mu_{E}$ and i=1,2,3.

#### 5.1.4. Stochastic Model of T Cell Dynamics in Homeostasis and during Infection

The rates introduced in Section 5.1.1, Section 5.1.2 and Section 5.1.3 for the nine different types of events (death, division and differentiation) allow us to define a stochastic (Markov) process to study the dynamics (or time evolution) of the T cell populations during homeostasis and viral infection [71]. We note that when a death event occurs, its corresponding population goes down by one cell. For instance, a death event for naive cells of clonotype 3 implies $n_{3}$ becomes $n_{3}-1$, after the death event takes place. If it is a division event, its population goes up by one cell (a division event for memory cells of clonotype 2 implies $m_{2}$ becomes $m_{2}+1$), and if the event is a differentiation, one population goes down by one and the other one up by one, with no change in the total population; that is, if the event is a differentiation in clonotype 1 from memory to effector, state $e_{1},m_{1}$ becomes $e_{1}+1,m_{1}-1$. These rules for the allowed transition events can be formally implemented as a multi-variate stochastic process, as defined in Ref. [71], which will determine how the T cell populations evolve in time due to homeostasis and during the two viral infections.

In the next section, we consider different hypotheses for focused and unfocused cross-reactivity [1], as introduced in Section 4, so that together with the dynamic rules provided in this section, we can explore what different bipartite recognition network structures imply for a given T cell clonotype. For unfocused cross-reactivity we consider a bipartite network where edges are added following a series of Bernoulli experiments, as described in Section 4.1.1. In the case of focused cross-reactivity, we consider two different types of networks: fixed degree focused and preferential attachment focused, as defined in Section 4.1.2.

### 5.2. Dynamics of T Cell Responses and Cross-Reactivity: Three networks

We now bring together the bipartite recognition networks introduced in Section 4, with the rules for division, differentiation and death of the naive, memory and effector T cell populations in health and disease defined in Section 5.1. We make use of the Gillespie algorithm to simulate how a finite set of clonotypes would evolve in time during homeostasis and during two different perturbations due to heterologous viral infections [82,83]. For the purposes of illustration and exposition, we only consider three different clonotypes (i=1,2,3) and a set of 18 pMHC complexes (nine for each viral infection considered). Thus, our bipartite network had three different TCR nodes and 18 VDP nodes. Three different sampling strategies, defined in Section 4, were used to create the recognition networks. We followed the T cell populations for a period of one year. Each infection lasted one week, and we assumed a six month period between infections. At time t=0, each clonotype consisted of five naive T cells only. Thus, there were no memory or effector cells of any TCR specificity to start with.

We can separate the parameters in Table 2 into three groups: parameters used for the construction of the VDP recognition network (*p*, *k*); parameters used for modeling the homeostatic dynamics ($\varphi_{i}$, $p_{i}$, $p_{ij}$, $p_{ijk}$, $\sigma_{M}$, $\mu_{N}$ and $\mu_{M}$); and finally, parameters used to model viral infection (β, γ(v), $\alpha_{N}$, $\alpha_{M}$, $\lambda_{E}$ and $\mu_{E}$). The values for the first group were chosen so that given our relatively small number of clonotypes and VDPs (3 and 18, respectively) we could construct networks that displayed some degree of cross-reactivity. For the second group, homeostatic parameters were selected so that the competition for survival stimulus from self-pMHC complexes in homeostasis did not favor a clonotype over others. Finally, for the third group the values of the parameters were chosen such that the stimulus provided by VDPs was several orders of magnitude larger than that provided by self-pMHCs (in homeostasis); that is, $\varphi_{i}\ll \gamma \left(v\right)$ and $\sigma_{M}\ll \alpha_{M}\gamma \left(v\right)$. We also assumed that a small fraction of effector cells, β, differentiate into memory after the infection is cleared [63]. For the death rates, we have assumed that $\mu_{M}<\mu_{N}<\mu_{E}$; that is, effector cells died at the highest rate, followed by naive and then memory T cells.

In Figure 10 we see (left panel) the bipartite recognition network used for the unfocused cross-reactivity hypothesis (as described in Section 4.1.1) with the parameters given in Table 2. The middle and right panels also show the results of a single stochastic simulation of a heterologous infection. Here, we can observe that the initial response is dominated by the second clonotype, and the remaining ones expand to a lesser degree. In this figure we also show the behavior of each of the separate pools (phenotypes) of cells from clonotype 1—that is, naive, effector and memory populations of cells belonging to clonotype 1.

From the results of this realization (see Figure 10), we see that during the initial challenge, the population of naive cells was not completely depleted as they differentiated into effector cells, allowing the naive pool to recover to homeostatic levels after the first infection was cleared. If we focused on effector cells, we see that during infection, the effector population expanded, and it contracted once the infection was cleared. Finally, we observed that memory cells were generated after the first infection was cleared. Once the second infection ended, there was an increase in this population, which was then homeostatically maintained by cytokine signals.

In Figure 11 we show the bipartite recognition network and the realization for the focused cross-reactivity hypothesis with a fixed degree, and in Figure 12 we show the results for the preferential attachment cross-reactivity hypothesis. For both hypotheses we made use of the parameters given in Table 2. In each instance, we observed a similar behavior to that of the unfocused hypothesis. T cell responses are dominated by one of the clonotypes, which is not necessarily the same for each infection.

From these initial and exploratory results, we conclude that the three different hypotheses of VDP recognition considered in this review lead to immune dynamics and clonal behavior that are immunologically plausible and realistic. For instance, in a given response to a viral infection, there is a dominant clonotype expanding to a greater degree than the rest of the clonotypes (or immunodominant clonotype), and thus, driving the dynamics of the immune response. Yet, the existence of focused and preferential cross-reactivity, as shown in Figure 12, might lead to greater competition between immunodominant clonotypes.

As it is essential to understand which cross-reactivity patterns are explored in nature, we made use of mathematically generated hypotheses together with current immunological evidence to test them with experimental and clinical datasets of T cell responses, which include quantitative measures of TCR clonotypes, as carried out in Refs. [22,33,85,86,87]. Network representations, as described in this section, can be used to define distances in the space of TCRs, based on their recognition profiles of peptides, or conversely we can define a distance between peptides based on which TCRs can recognize them [88,89,90,91]. However, these definitions of distance rely on having previous and precise knowledge of which peptides will be recognized by a given TCR (since we need to construct the bipartite recognition network before we can calculate any distance between nodes of the same family). This is of course, not information that we have a priori for most TCRs and pMHCs. For this reason, previously defined distances between TCRs are guided by structural information on their pMHC binding properties. For example, TCRdist is based on the structure of the CDR1, CDR2 and CDR3 loops [33]. In Section 6 we put forward a different mathematical approach to characterizing and quantifying TCR-pMHC recognition. This approach has a natural distance encoded and is explored in what follows.

## 6. Modeling T Cell Cross-Reactivity with a Distance
in Epitope Space

In our mathematical models, a clonotype is a set of T cells that have the same pattern of recognition in the space of epitopes (pMHC complexes), because they share the same TCR. In the broadest definition, any clonotype whose cells recognize more than one epitope would be described as cross-reactive. However, the term is more usefully reserved for clonotypes that are parts of immune responses to different pathogens [15,16].

Lythe et al. constructed a computational model of T cell repertoire homeostasis based on the hypothesis that every TCR, independently, has a fixed probability, *p*, of recognizing a given epitope [65]. The value of *p*, sometimes called the precursor frequency, can be estimated in various ways from experimental data [92,93], notably by using tetramers to extract epitope-specific T cells from laboratory mice [2,94]. A few hundred epitope-specific cells are typically counted, out of tens of millions of naive T cells in one mouse, suggesting that *p* is in the range $10^{-6}-10^{-5}$ [2,94]. The number of (mostly self) epitopes in a mouse or human body, *M*, is estimated to be many millions [15,16,65]. Thus, every TCR clonotype is expected to be cross-reactive in the sense that the average number of epitopes it recognizes, pM, is much greater than 1 [15,16].

Does a repertoire in which cells of a typical TCR clonotype recognize many different epitopes display cross-reactivity? To consider this, we estimated how many T cells in a mouse would recognize two distinct epitopes. If the probability of recognition of each epitope is *p*, independently, then the probability that a given TCR recognizes both epitopes is $p^{2}$, which is smaller than $10^{-10}$. That is, fewer than one TCR in $10^{10}$ would, by chance, recognize both epitopes. Thus, even if a mouse’s repertoire contains more than $10^{7}$ different TCRs, it is unlikely to possess even a single such functionally cross-reactive cell. To construct a repertoire with functional cross-reactivity, we must assume that a TCR that recognizes one influenza epitope is likely (that is, more likely than a randomly-chosen TCR) to recognize a "similar" epitope. Thus, we must introduce a notion of distance between epitopes, so that we can say which epitopes are close to each other (similar) and which are not. We describe this notion in what follows, and a mathematical model to define the dynamics of T cell clonotypes.

In our model, the homeostasis of a TCR repertoire can be summarized as follows. New T cell clonotypes emerge from the thymus at a rate θ with a size of $n_{0}$ cells per clonotype. Every T cell independently has death rate μ. That is, the mean lifetime of a cell is 1/μ and does not depend on its TCR. A TCR clonotype is defined by its pattern of recognition in the space of epitopes (self and viral pMHC complexes). Two epitopes are close to each other if they are similar in their molecular structures, which could in principle be represented in a high-dimensional space. Here, we construct a simple example: we imagine the space of epitopes to be a circle as shown in Figure 13. No part of epitope space is privileged, yet a sense of distance between epitopes is introduced. One epitope is represented as a point on the circle and one clonotype is an arc, covering a fraction, *p*, of the circle (see Figure 13). Cells of a clonotype recognize all of the epitopes in the arc. There are *M* epitopes randomly placed on the circle, and N(t) is the number of T cell clonotypes at time *t*. Each new clonotype’s arc is centered at a random point on the circle. An illustration of the resulting clonotype dynamics, in homeostasis and during an infection, is given in the Appendix A.

In homeostasis, each self-pMHC provides stimulus to T cells at a low rate. An episode of infection is modeled as a short period during which the division stimulus from one epitope (we assume one epitope per infection) is large. Hence, the number of cells in each of the clonotypes which recognize this epitope increases rapidly during its infection period. We model this as follows. If clonotype *i* has $n_{i}\left(t\right)$ cells at time *t*, then the probability that one of them dies in the interval (t,t+Δt) is $\mu n_{i}\left(t\right)\Delta t$, in the limit $\Delta t\to 0^{+}$. That is, $P\left(n_{i}\left(t+\Delta t\right)=n_{i}\left(t\right)-1\right)\;=\mu n_{i}\left(t\right)\Delta t$. The probability that one cell of clonotype *i* divides between *t* and t+Δt is equal to $\Lambda_{i}\left(t\right)n_{i}\left(t\right)\Delta t$, where
(19)$$\Lambda_{i}\left(t\right)=\sum_{q\in A_{i}}\frac{\gamma_{q}}{N_{q}\left(t\right)},$$

$A_{i}$ is the set of epitopes in clonotype *i*’s arc, and $\gamma_{q}$ is the stimulus rate to divide provided by epitope *q*. At t=0, $\gamma_{q}=\gamma$ for all *q*. $N_{q}\left(t\right)$ is the number of cells at time *t* that recognize epitope *q*.

The number of T cells that recognize a given epitope changes with time as cells die and divide. Homeostasis, in mice and in humans, may be compared using the parameter $\alpha =\frac{M\gamma }{\theta n_{0}}$, which is the ratio of the number of new cells, per day, produced by division in the periphery to the number of new cells released from the thymus to the periphery [65,66]. In mice, α≪1, and in humans α>1 [65,66,95,96]. In Figure 14 and Figure 15, we choose α=0.1. In homeostasis, the mean total number of cells is given by $S=(\theta n_{0}+M\gamma )/\mu$ and the mean number of clonotypes by $N^{*}≃S\left(0.433-\log(n_{0}\alpha )\right)$ [65,66].

The computational framework introduced in this review can be used to construct simple conceptual models, with a few hundred clonotypes, and larger-scale simulations of a mouse, or human, T cell repertoires. Scaling up or down is possible, based on the choice of model parameters. In Table 3 we compare the parameter values used in the numerical realizations of Section 6 with conjectured values in mice and humans, for naive CD4${}^{+}$ T cells [6,97,98]. The most important parameter distinguishing mice from humans is α, which measures the strength of peripheral division to thymic export [95]. The parameter values were assumed to be constant here; in reality they depend on age due to the slow decline of thymic production [66,99,100].

### Two Episodes of Infection

Infection is modeled as a short period during which the division stimulus from one viral epitope (we assumed one epitope per infection) is large; hence the number of cells in each of the clonotypes which recognizes this epitope will increase rapidly. The epitope *e* responsible for the infection is present for a time τ=0.5 month (two weeks), with $\gamma_{e}=K\gamma$ and K≫1. When two epitopes are far apart in the space (of epitopes), as in Figure 14, the two episodes of infection have little influence on each other. In each case, the number of epitope-specific T cells increases rapidly and a smaller effect on the number of clonotypes is also seen.

In Figure 15, epitopes $e_{1}$ and $e_{2}$ are similar enough—illustrated by the blue and red dots on the circle—that there are T cells that recognize both. This cross-reactive population is boosted by the first infection; declines between the end of the first and start of the second infection; and then, is boosted to a higher peak during the second infection. The higher peak is entirely due to cross-reactivity: a distinguished memory T cell phenotype was not included in this model. In other words, the repertoire’s response to the second infection is boosted by the long-term effects of the response to the previous infection.

## 7. Discussion and Conclusions

While much has yet to be determined about the nature of T cell cross-reactivity, it is clear that cross-reactive CD8${}^{+}$ T cells play a role in the immune response to respiratory infections, particularly those that an individual can experience repeatedly during a life time. This is most clearly exemplified at the moment by influenza and coronaviruses. The current pandemic has placed common cold coronavirus infections in the spotlight because of the potential for cross-reactive T cell immunity, and thus, a likely source of protection from SARS-CoV-2. While much remains to be learned about CD8${}^{+}$ T cell cross-reactivity and SARS-CoV-2, recent findings of cross-reactive CD4${}^{+}$ T cells are encouraging [8]. With an epitope-specific approach, a clearer picture may emerge as immunodominance hierarchies for SARS-CoV-2 CD8${}^{+}$ epitopes are determined. Cross-reactive T cells may also play an important role in a future universal influenza vaccine. In addition, deciphering novel aspects and the rules of cross-reactivity may prove to be an important tool for understanding T cell immune responses to other viral infections and autoimmune diseases.

In Section 4, Section 5 and Section 6, we introduced two different approaches to quantify cross-reactivity, using bipartite recognition networks and a simple epitope space. The first allowed us to compare unfocused and focused cross-reactivity, and thus propose different solutions for the recognition of VDPs and TCR clonotype clustering (Section 4 and Section 5). The second approach led to the consideration of cross-reactivity as a measure of the overlap between TCR clonotypes that recognize similar (and thus, close in epitope space) epitopes (see Section 6). Mathematically generated hypotheses can then be compared with current immunological evidence [22,33,85,86,87]. In this way, quantitative exploration of immune exposures, associations between MHC alleles and shared TCRs in large human cohorts could inform model development and validate the hypotheses being put forward [22] (see Figure 3).

As discussed in this review, mathematical methods and models have already been used in the context of influenza virus replication in cells [37,38], and TCR diversity [33,101]. More recently, mathematical approaches have been used to model exposure of healthcare professionals to SARS-CoV-2 and norovirus in the context of hospital patient rooms [102], highlighting the importance and usefulness of mathematical modeling in infection settings. We hope this review will pave the way to greater joint experimental and theoretical efforts in the area of T cell cross-reactivity in health and disease.

## Figures and Tables

**Figure 1 viruses-13-01786-f001:**
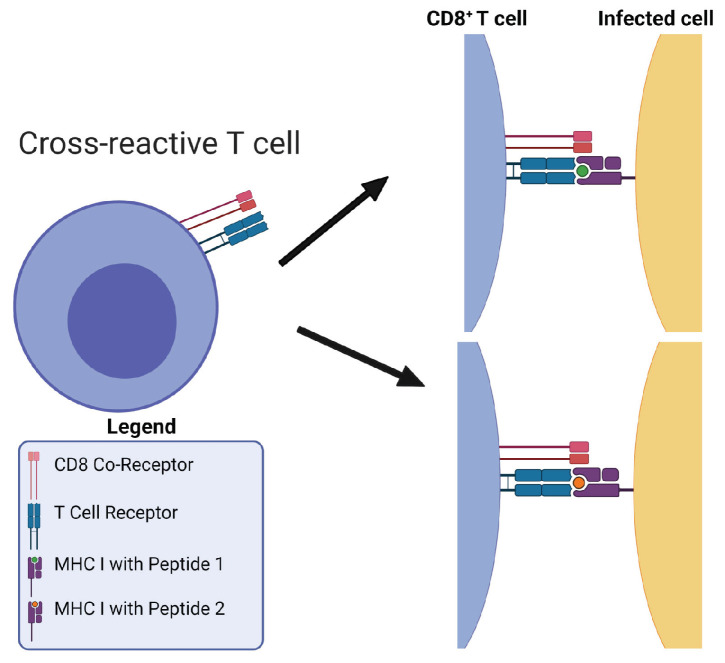
Definition of TCR cross-reactivity. Presentation of two distinct epitopes in the context of MHC-I and the recognition of both of these epitopes by a single cross-reactive TCR on a CD8${}^{+}$ T cell.

**Figure 2 viruses-13-01786-f002:**
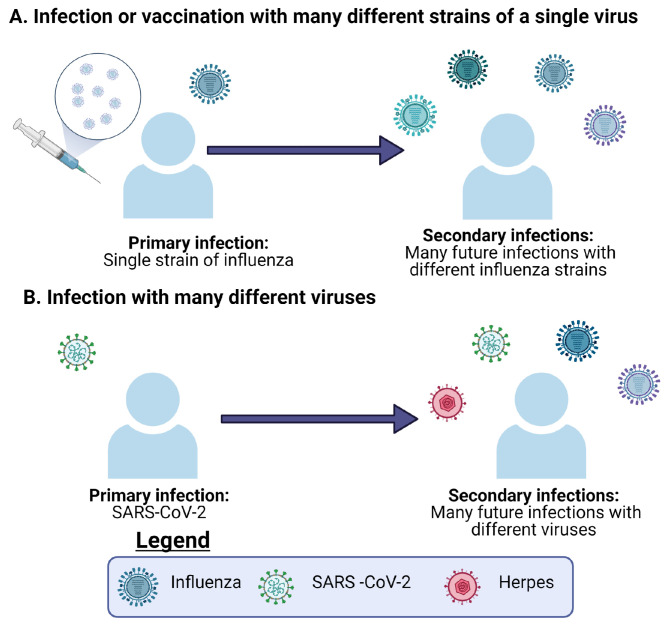
The definitions of heterologous infection and heterologous immunity. (**A**) An individual infected with several heterologous influenza viruses. (**B**) An individual infected with different viruses. In both (**A**,**B**), the primary infection or vaccination serves to generate immunity, which is then considered pre-existing when the individual is later infected with different viruses.

**Figure 3 viruses-13-01786-f003:**
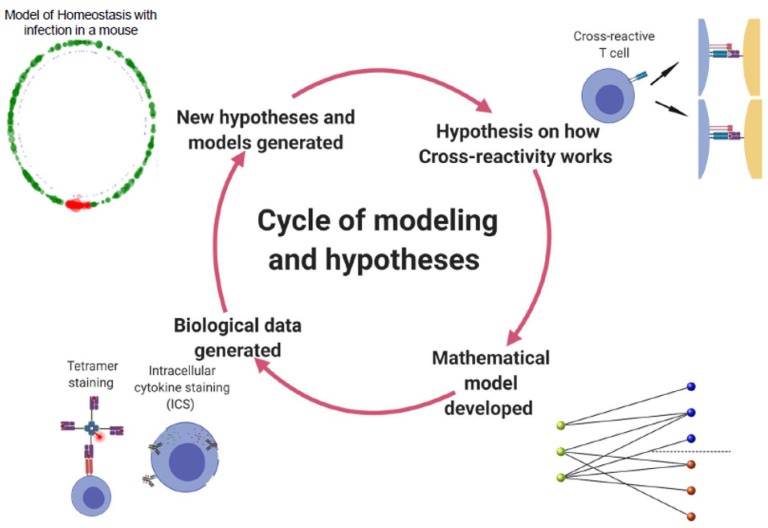
The cycle of mathematical modeling and the investigation of TCR cross-reactivity.

**Figure 4 viruses-13-01786-f004:**
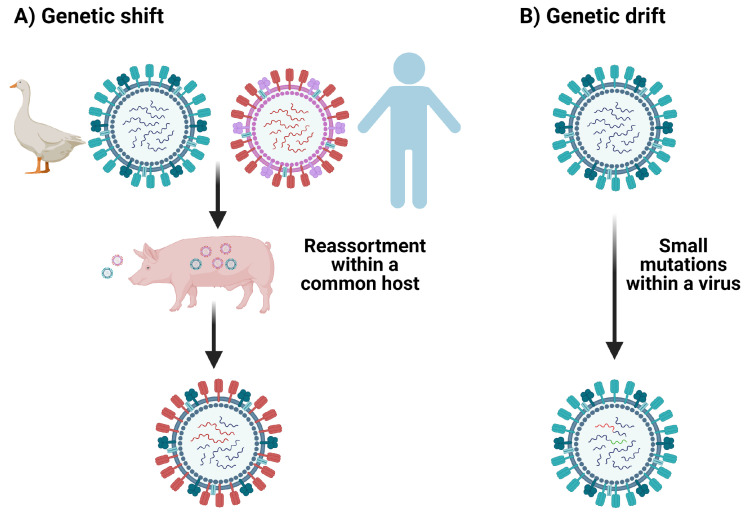
Mechanisms of mutation of influenza viruses. (**A**) The swapping of gene segments between influenza viruses via reassortment in a common host, illustrating genetic shift. (**B**) Small mutations gained from error-prone replication illustrating genetic drift.

**Figure 5 viruses-13-01786-f005:**
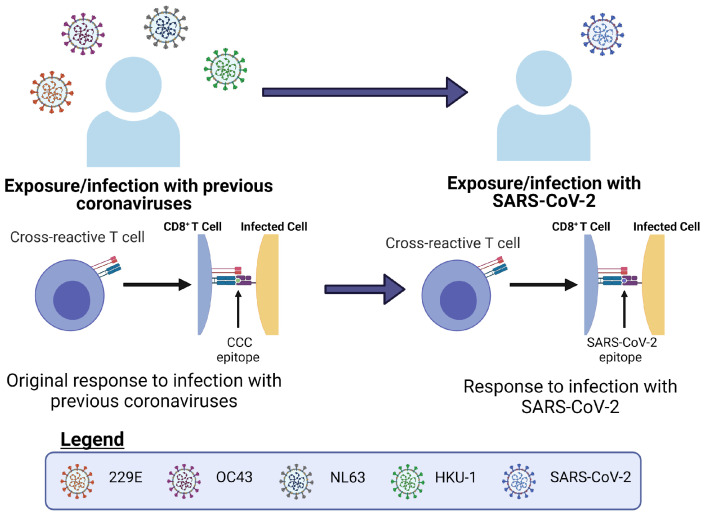
Individuals with previous exposure to common cold coronaviruses (CCCs) most likely have cross-reactive pre-existing immunity to pandemic SARS-CoV-2.

**Figure 6 viruses-13-01786-f006:**
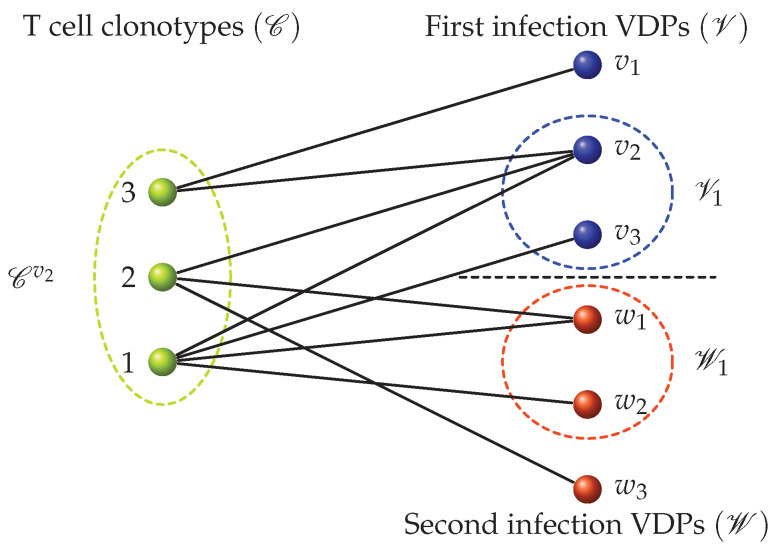
Bipartite recognition network generalizing the model developed by Stirk et al. [56,57,58,59,60]. Each green circle (or node) represents a T cell clonotype. Each blue or red circle represents a VDP node from a first and second infection, respectively. An edge between a green and a blue or red node represents the ability of that clonotype to recognize the VDP it is connected to. For any VDP, *v*, the set of clonotypes that can recognize it is denoted by $C^{v}$, and for a given clonotype, *i*, the set of first infection VDP it can recognize is denoted by $V_{i}$, and the set of second infection VDPs it can recognize is denoted by $W_{i}$. For example, in the bipartite recognition network chosen, the set $C^{v_{2}}$ consists of clonotypes 1, 2 and 3; the set $C^{w_{2}}$ consists of clonotype 1; the set $V_{3}=\left\{v_{1},v_{2}\right\}$; the set $W_{2}=\left\{w_{1},w_{3}\right\}$; and the set $W_{3}=\left\{Ø\right\}$.

**Figure 7 viruses-13-01786-f007:**
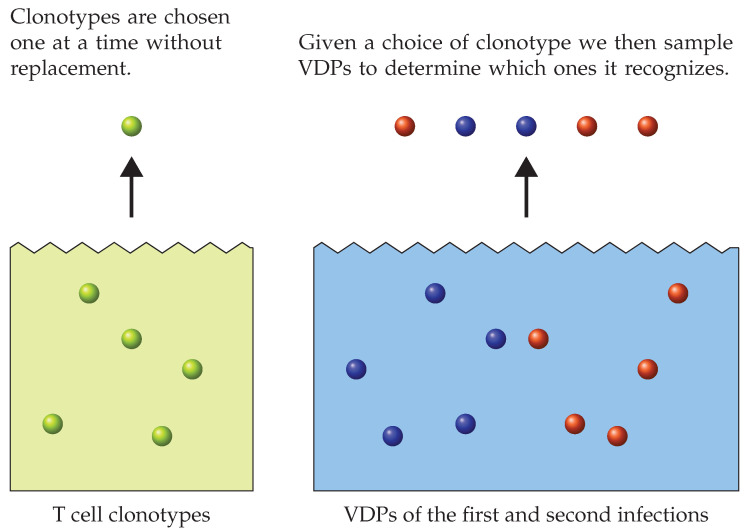
(**Left**) Bag of T cell clonotypes. We chose clonotypes at random from the bag, one at a time and without replacement. (**Right**) A bag of all the available VDPs (from the first and second infection). Once we chose a T cell clonotype, we then sampled VDPs. The sampled VDPs were assigned edges to the chosen clonotype. By considering different sampling strategies from the bag of VDPs, we generated distinct bipartite recognition networks with different properties, such as degree of clustering or cross-reactivity [1,13].

**Figure 8 viruses-13-01786-f008:**
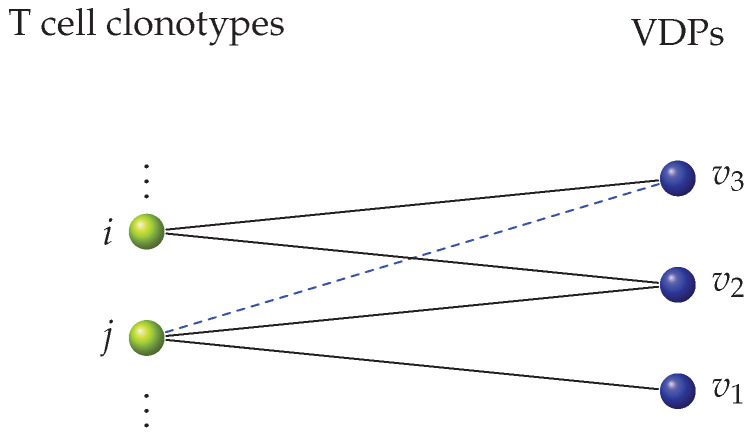
Preferential attachment network: a second way to define a focused cross-reactivity network. If two clonotypes share a VDP, we hypothesized that their TCR structures might be similar, and thus, that they might share other VDPs as well (with a greater probability than by chance). Clonotypes *i* and *j* both recognize $v_{2}$ and clonotype *i* recognizes $v_{3}$. Hence, we sampled from the set of VDPs recognized by clonotype *i* (and not in the set of VDPs recognized by clonotype *j*). In this case this set was $v_{3}$, so we sampled again to decide whether *j* can also recognize $v_{3}$.

**Figure 9 viruses-13-01786-f009:**
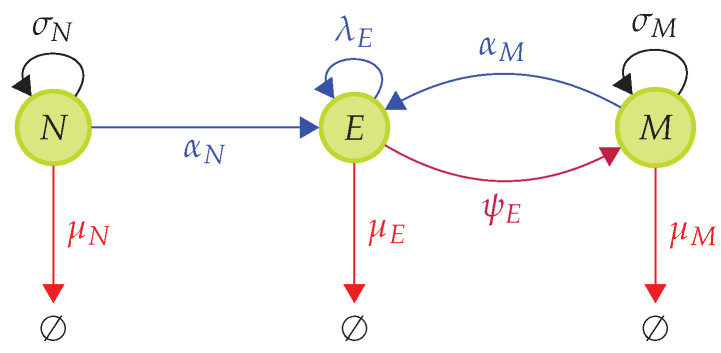
Differentiation pathway during a T cell immune response to a viral infection. During a viral infection we have the following events: (i) Naive cells, *N*, differentiate into effector cells with rate $\alpha_{N}$ when presented with VDPs they recognize. (ii) Effector cells, *E*, proliferate with rate $\lambda_{E}$. (iii) Memory cells, *M*, differentiate into effector cells with rate $\alpha_{M}$ when presented with VDPs they recognize. Naive cells are homeostatically maintained by survival stimuli from self-pMHCs with rate $\sigma_{N}$. Memory cells are homeostatically maintained by survival stimuli from cytokines with rate $\sigma_{M}$. Naive, effector and memory cells die at (per cell) rates $\mu_{N}$, $\mu_{E}$ and $\mu_{M}$, respectively. Once infection is cleared, effector cells differentiate into memory cells at a rate $\psi_{E}$. We assume $\psi_{E}$ is such that $\frac{\psi_{E}}{\psi_{E}+\mu_{E}}=\beta$, with β a fraction in the range of 5–10 % [63].

**Figure 10 viruses-13-01786-f010:**
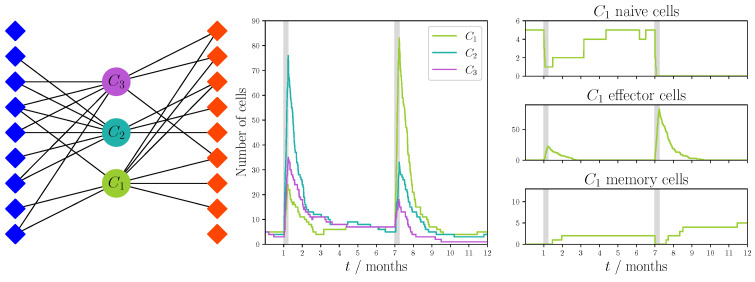
Single realization of an immune response to two viral challenges with an unfocused recognition network and dynamics defined in Section 5.1. The network was constructed using the sampling strategy described in Section 4.1.1 with probability of success p=8/18. Blue diamonds represent VDPs present during the first viral challenge and red diamonds represent those present during the second challenge. We note that not every VDP was recognized by the clonotypes, since the samples were drawn in a random fashion, and there was always a probability 1−p of no recognition between the chosen clonotype and a VDP. Model parameters are described in Table 2. The python code used to generate this figure has been made available at https://doi.org/10.5281/zenodo.5227343 [84].

**Figure 11 viruses-13-01786-f011:**
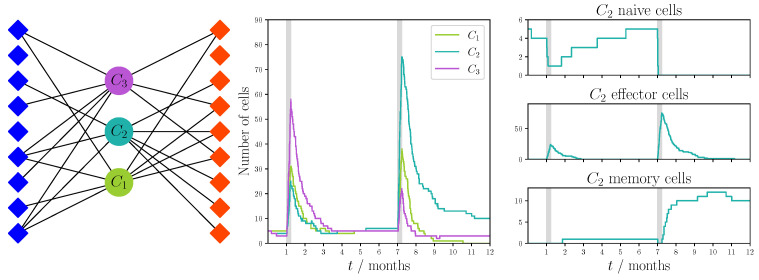
Single realization of an immune response to two viral challenges with a focused fixed degree recognition network and dynamics defined in Section 5.1. The network was constructed using the sampling strategy described in Section 4.1.2 with degree k=8. Blue diamonds represent VDPs present during the first viral challenge and red diamonds represent those present during the second challenge. We note that not every VDP is recognized by the clonotypes. Model parameters are described in Table 2. The python code used to generate this figure has been made available at https://doi.org/10.5281/zenodo.5227343 [84].

**Figure 12 viruses-13-01786-f012:**
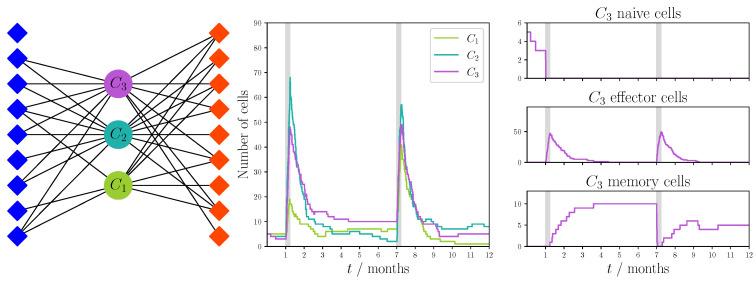
Single realization of an immune response to two viral challenges with a focused preferential attachment recognition network and dynamics defined in Section 5.1. The network was constructed using the sampling strategy described in Section 4.1.2 with probability of success p=8/18. Blue diamonds represent VDPs present during the first viral challenge and red diamonds represent those present during the second challenge. We note that not every VDP was recognized by the clonotypes. Model parameters are described in Table 2. The python code used to generate this figure has been made available at https://doi.org/10.5281/zenodo.5227343 [84].

**Figure 13 viruses-13-01786-f013:**
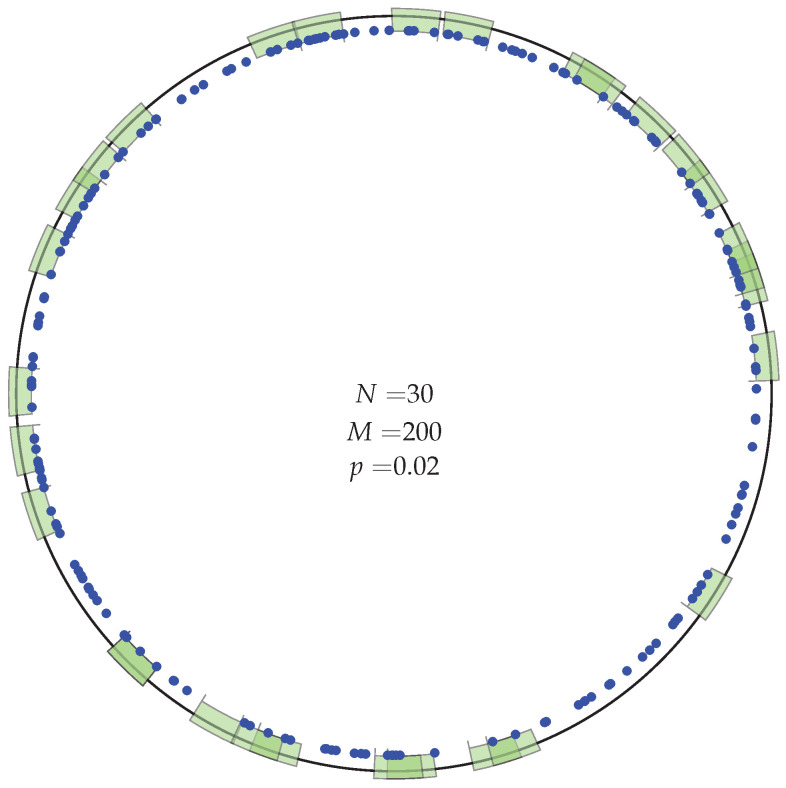
An example of epitope space: the unit circle. Each blue dot is a pMHC complex, *M* of which are scattered randomly on the circle. Each green arc is the recognition region of a TCR clonotype. We assume each TCR clonotype occupies a fraction *p* of the unit circle. The centers of the arcs are scattered randomly on the circle. Arcs may overlap: pMHCs in the overlap region are recognized by all the TCRs defined by those arcs.

**Figure 14 viruses-13-01786-f014:**
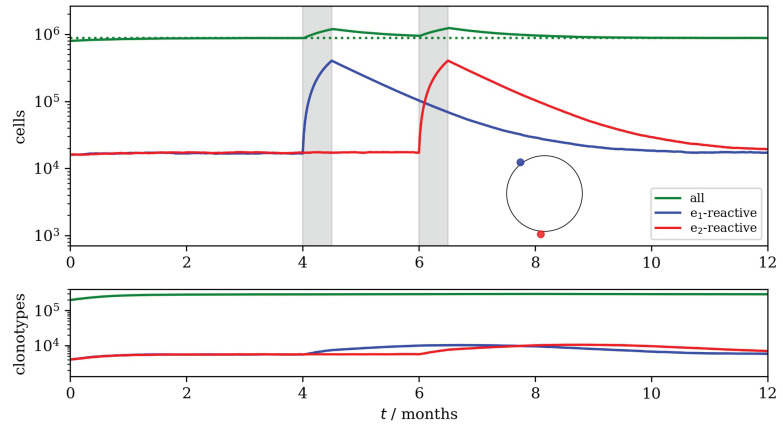
Two infection episodes: different viral epitopes. The blue and red dots on the circle indicate the locations of the two different viral epitopes considered, $e_{1}$ (blue) and $e_{2}$ (red). The two shaded intervals indicate infection with epitopes $e_{1}$ and $e_{2}$, respectively. Top: The total number of cells is shown in green. The blue and red lines show the numbers of cells recognizing $e_{1}$ and cells recognizing $e_{2}$, respectively. Bottom: The total number of surviving clonotypes as a function of time is shown in green. The blue and red lines show the numbers of surviving clonotypes recognizing $e_{1}$, and of surviving clonotypes recognizing $e_{2}$, respectively. Parameter values were: μ=1 month^−1^, $\theta =10^{5}$ month^−1^, $\gamma =4\times 10^{2}$ month^−1^, $n_{0}=8$, $M=2\times 10^{2}$, p=0.02 and $K=10^{3}$.

**Figure 15 viruses-13-01786-f015:**
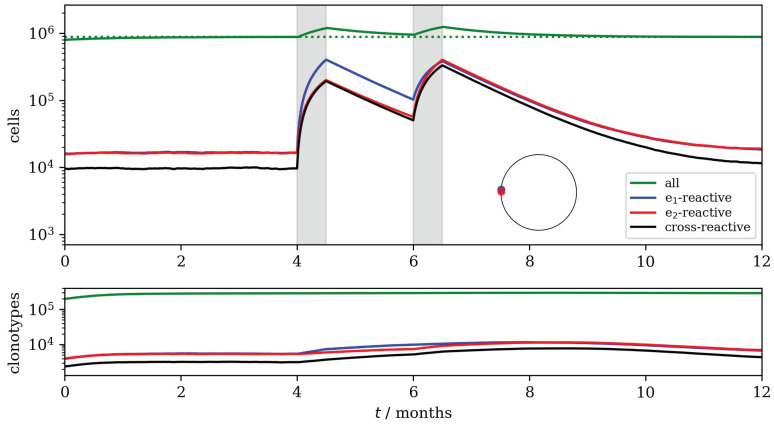
Two infection episodes: similar viral epitopes. The blue and red dots on the circle indicate the locations of $e_{1}$ and $e_{2}$, respectively. The two shaded intervals indicate infection with epitope $e_{1}$ and $e_{2}$, respectively. Top: The total number of cells is shown in green. The blue, red and black lines show the numbers of cells recognizing $e_{1}$; $e_{2}$; and both $e_{1}$ and $e_{2}$. Bottom: The total number of surviving clonotypes as a function of time is shown in green. The blue, red and black lines show the numbers of surviving clonotypes recognizing $e_{1}$; $e_{2}$; and both $e_{1}$ and $e_{2}$. Parameter values were: μ=1 month^−1^, $\theta =10^{5}$ month^−1^, $\gamma =4\times 10^{2}$ month^−1^, $n_{0}=8$, $M=2\times 10^{2}$, p=0.02 and $K=10^{3}$.

**Table 1 viruses-13-01786-t001:** Key sets and parameters used for the proposed model of the dynamics of T cells during a viral infection.

Description	Symbol
Set of clonotypes modeled	C
Set of clonotypes that can recognize VDP *v*	$C^{v}$
Set of VDPs presented during the first infection	V
Set of VDPs presented during the first infection recognized by clonotype *i*	$V_{i}$
Set of VDPs presented during the second infection	W
Set of VDPs presented during the second infection recognized by clonotype *i*	$W_{i}$
Probability of a random VDP to be recognized by a clonotype	*p*
Number of VDPs a clonotype can recognize	*k*
Total naive homeostatic stimulus rate for clonotype *i*	$\varphi_{i}$
Memory homeostatic division rate	$\sigma_{M}$
Effector to memory differentiation fraction	β
Stimulus rate provided by VDP *v*	γ(v)
Naive death rate (per cell)	$\mu_{N}$
Memory death rate (per cell)	$\mu_{M}$
Effector death rate (per cell)	$\mu_{E}$

**Table 2 viruses-13-01786-t002:** Parameter values to simulate the dynamics of three T cell clonotypes for a period of one year.

Description	Symbol	Units	Value
Probability of a VDP being drawn	*p*	-	8/18≈0.44
Degree of a clonotype in the bipartite network	*k*	-	8
Total naive homeostatic stimulus rate for clonotype *i*	$\varphi_{i}$	$\mathrm{cells}\times {\mathrm{year}}^{-1}$	10
Probability of not sharing self-pMHCs with other clonotypes	$p_{i}$	-	4/9
Probability of sharing self-pMHCs with one clonotype	$p_{ij}$	-	2/9
Probability of sharing self-pMHCs with two clonotypes	$p_{ijk}$	-	1/9
Memory homeostatic division rate	$\sigma_{M}$	${\mathrm{year}}^{-1}$	1
Effector to memory differentiation fraction	β	-	10%
VDP stimulus rate	γ(v)	$\mathrm{cells}\times {\mathrm{year}}^{-1}$	$10^{3}$
Naive to effector differentiation constant	$\alpha_{N}$	-	1
Memory to effector differentiation constant	$\alpha_{M}$	-	2
Effector division constant	$\lambda_{E}$	-	1
Naive death rate (per cell)	$\mu_{N}$	${\mathrm{year}}^{-1}$	1
Memory death rate (per cell)	$\mu_{M}$	${\mathrm{year}}^{-1}$	0.8
Effector death rate (per cell)	$\mu_{E}$	${\mathrm{year}}^{-1}$	20

**Table 3 viruses-13-01786-t003:** Parameters of the epitope distance model. The values used in Figure 14 and Figure 15 are compared with conjectured values in mice and humans. The thymus produces new clonotypes at rate θ, each starting with $n_{0}$ cells. The parameter α is the rate of cell division in the periphery, Mγ, divided by $\theta n_{0}$.

Parameter	Figures	Mouse	Human	Definition
θ	$10^{5}$ month^−1^	$10^{7}$ month^−1^	$10^{9}$ month^−1^	thymic output rate
$n_{0}$	8	8	8	thymic clonal size
μ	1 month^−1^	1 month^−1^	0.01 month^−1^	per cell death rate
α	0.1	0.1	10	peripheral division strength
*M*	200	$10^{12}$	$10^{13}$	total number of self-pMHCs
*p*	0.05	$10^{-5}$	$10^{-5}$	precursor frequency

## Data Availability

The python codes used to generate Figure 10, Figure 11 and Figure 12 are available at https://doi.org/10.5281/zenodo.5227343 [84].

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
