# Peer review of "Quantifying T Cell Cross-Reactivity: Influenza and Coronaviruses"

_viruses, 2021, doi:10.3390/v13091786_

Round 1
Reviewer 1 Report
The manuscript is well written, and the content will be interesting and beneficial in the research community.
Minor points:
- In figure 1, the colors of peptide 1 and peptide 2 seem to be the same. It could be better to change.
- There is no legend for nodes in diamond colored orange and blue in figures 10, 11, and 12. They may be VDP but it would be better to be noted in the figure legend.
- The equation on Page 14 Line 424, n = (n1, ...) may be the same as Eq. (1) on Page 12 but no mention of that. One can guess but it may be better to mention for increasing the readability.
- The python code available at the URL in Figure 10 may not work properly on macOS. Recognition-network-gillespie.py seems to work fine but Plots.py exited with an error message, RuntimeError: Failed to process string with tex because latex could not be found.
Reviewer 2 Report
The review Gaevert et al. is an interesting review paper that presents mathematical models for quantitification of T-cell cross-reactivity in coronavirus and influenza infections. The paper is well written and covers an interesting topic. I have only two minor issues to be addressed.
Major comment:
Title: Consider adapt the title to “Quantifying cross-reactivity in T-cells: influenza and coronaviruses”. The text focuses on T-cell cross-reactivity and not e.g. antibody cross-reactivity.
Minor comment:
Figure 3: font is too small within the figure. I can not read details, e.g. “New hyptoheses and modesl generated”.
Reviewer 3 Report
This review discussed studying pre-existing immunity with mathematical modeling methods. Honestly, many contents of modeling are out of my expertise. I tried my best to read and understand these studies. Here is my comments for your reference.
- The review discussed a very important topic: pre-existing immunity. Section 2 and 3 that review the T cell cross-reactivity in Influenza viruses and coronaviruses is a good update of T cell cross-reactivity.
- Section 4-6 discussed the modeling of T cell cross-reactivity with different dynamics of T cell responses: to different pathogens, in viral infections, and with a distance in epitope space. Even though I do not understand the math modeling part, but I think these three considerations are well-designed, and represent the immunity scenarios in the viral infection. The only question I have is that the authors used many " we ..." in these sections with the references authored with different researchers that are not the authors of this review, Were these works all did by the same group? I also have a hard time figuring out which figures or tables are published or unpublished. I would suggest that the author cite the reference on each previous published figure and table and indicated the unpublished figure as "unpublished"
- some minor typos and writing issues:
- line19: typo: should be 1012?
- Line 231: what did the large CD8+ T cell activation" mean? Did you mean "a large amount of cell"
